# Scaling behavior of electron decoherence in a graphene Mach-Zehnder interferometer

M. Jo[1,5], June-Young M. Lee [2,5], A. Assouline[1], P. Brasseur[1], K. Watanabe [3], T. Taniguchi [3], P. Roche [1], D. C. Glattli [1], N. Kumada [4], F. D. Parmentier [1], H. -S. Sim [2] ✉ & P. Roulleau [1] ✉

Over the past 20 years, many efforts have been made to understand and control decoherence in 2D electron systems. In particular, several types of electronic interferometers have been considered in GaAs heterostructures, in order to protect the interfering electrons from decoherence. Nevertheless, it is now understood that several intrinsic decoherence sources fundamentally limit more advanced quantum manipulations. Here, we show that graphene offers a unique possibility to reach a regime where the decoherence is frozen and to study unexplored regimes of electron interferometry. We probe the decoherence of electron channels in a graphene quantum Hall PN junction, forming a Mach-Zehnder interferometer[1,2], and unveil a scaling behavior of decay of the interference visibility with the temperature scaled by the interferometer length. It exhibits a remarkable crossover from an exponential decay at higher temperature to an algebraic decay at lower temperature where almost no decoherence occurs, a regime previously unobserved in GaAs interferometers.

The field of electron quantum optics relies on the analogy between the propagation of electrons in a quantum conductor and that of photons in quantum optics experiments. This research field emerged in the late nineties with the possibility of manipulating electron beams in condensed matter systems while preserving their wave-particle nature. It has proven since then to grant a fundamental understanding of quantum electronics down to the single-particle excitation. The prototypical systems of electron quantum optics are two-dimensional conductors in the quantum Hall effect regime. This regime is reached under strong perpendicular magnetic field and is characterized by the existence of one-dimensional, chiral and dissipationless electronic channels propagating along the edges of the sample. Those quantum Hall edge channels can be directly viewed as the analog of optical fibers for electrons. A large majority of these experiments has been performed in GaAs/AlGaAs semiconductor heterostructures, where it has been shown that decoherence can stem from different sources: edge

reconstruction by disorder[3], *intra-channel* Coulomb interactions within a single edge channel[4,5], and *inter-channel* Coulomb interactions between adjacent edge channels[6–8]. The latter mechanism is thought to be the main hindrance in realizing complex quantum circuits with quantum Hall edge channels in GaAs, and has received considerable theoretical and experimental attention. In particular, recent experiments have shown that it can be diminished by a somewhat cumbersome engineering of the edge channels[9,10]. Despite this, identifying decoherence sources remains an open problem, with e.g. several previously overlooked dissipation mechanisms that have been put into light in the last few years[11,12], and experimental observations that are still debated after more than a decade[13,14]. Probing decoherence in quantum Hall edge channels realized in different 2D materials potentially allows tackling this issue, by providing an apparently similar system whose intrinsic parameters (e.g. electron velocity, capacitive coupling and screening, geometry) are nonetheless sufficiently

[1]SPEC, CEA, CNRS, Université Paris-Saclay, CEA Saclay, 91191 Gif sur Yvette, Cedex, France. [2]Department of Physics, Korea Advanced Institute of Science and Technology, Daejeon 34141, Korea. [3]National Institute for Materials Science, 1-1 Namiki, Tsukuba 305-0044, Japan. [4]NTT Basic Research Laboratories, NTT Corporation, 3-1 Morinosato-Wakamiya, Atsugi 243-0198, Japan. [5]These authors contributed equally: M. Jo, June-Young M. Lee. ✉e-mail: hs_sim@kaist.ac.kr; preden.roulleau@cea.fr

different to obtain a full picture of decoherence. In this letter, we probe the decoherence in an electronic Mach-Zehnder interferometer realized in a graphene PN junction in the quantum Hall regime, in which the typical energy scales for decoherence are an order of magnitude larger than in GaAs/AlGaAs. This allows us to observe for the first time a remarkable universal behavior in the dependence of the interferences with the temperature. Together with the bias voltage dependence of the visibility, we show that for those systems decoherence is mainly due to *intra-channel* Coulomb interactions.

## Results

In our PN junction, a graphene monolayer is encapsulated by hexagonal boron nitride layers, and the density of the left and right halves are controlled independently using bottom and top gates, where the former (resp. latter) is covering the whole sample (resp. only the right half), as shown in Fig. 1a and the "sample description" section in the Supplementary Material. Under a perpendicular magnetic field, the left half becomes a P region of filling factor $v_p = -1$. Along the boundary of the P region, a spin-up channel circulates clockwise. The right half is an N region of $v_n = 2$. Along its boundary, two channels having opposite spin circulate counterclockwise. As a result, the junction interface has the three co-propagating channels. The two spin-up channels have opposite valley-isospin[1]. A Mach-Zehnder interferometer is formed at the PN interface by applying the top and bottom side gates (see Fig. 1). Along the top edge, the injected current $I_0$ is carried by the two edge channels of the N region. Half of the current, resulting from spin down carriers, cannot flow to the P region, because of large energy cost for spin flip. The other half $I_0/2$ with spin up carriers, on which we focus

hereafter, can contribute to the transmitted current $I_T$. Therefore the transmission probability was measured as $T_{MZ} = I_T / (I_0/2)$. The filling factors $v_1$ and $v_2$ below the side gates are controlled independently. In the "large" interferometer, $v_{i=1,2} = -1$ (see the first panel of Fig. 1b). In this case, the spin-up channels from the P and N regions collide at the top and bottom "edge" intersections of the PN interface with the graphene edge below the side gates, leading to formation of their beam splitters (which are called valley splitters[2]). The atomic structure of the graphene edge causes sharp potential change at the edge intersection, hence, scattering between the spin-up channels having opposite valley-isospin[15,16]. The two spin-up channels co-propagating along the PN interface and their beam splitters at the top and bottom edge intersections constitute the large interferometer that exhibits transmission oscillations of period $\Delta B = 20$ mT as a function of the magnetic field (note that this period is slightly different from the one reported in ref. 2 although it is the same device. Between the two measurements, the experimental setup has been deeply modified. It also corresponds to two different cooldowns with different gate voltages and edge electrostatics). The arm length of the large interferometer is estimated as $L = 1.5$ μm from the sample geometry (see the "sample geometry" section in the Supplementary Material). When a side gate is further tuned to have $v_1 = 0$ or $v_2 = 0$, the Aharonov-Bohm oscillations disappear as reported in refs. 1, 2, but they reappear in certain ranges of the side gate voltage (see the "tuning length of the interferometer" section in the Supplementary Material). Two different periods $\Delta B = 34.5$ mT and 81 mT of the reappeared Aharonov-Bohm oscillations are observed (see the second and third panels of Fig. 1b), and they are much larger than the period $\Delta B = 20$ mT of the large

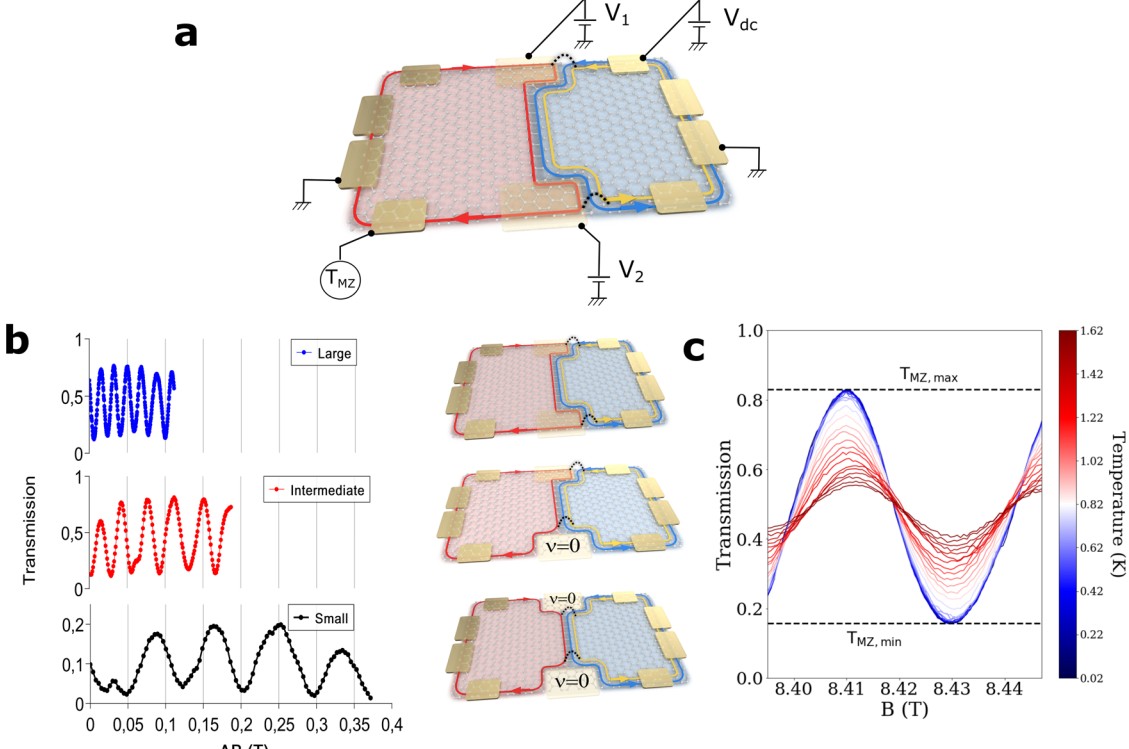

**Fig. 1 | Experimental setup and Aharonov-Bohm oscillations. a** Schematic representation of the PN junction. The N region is depicted in blue, the P one in pink. Electrons are injected from the upper right ohmic contact by applying a bias voltage $V_{DC}$, and electron transmission $T_{MZ}$ is measured at the lower left contact. **b** Left panel: $T_{MZ}$ with respect to change $\Delta B$ of the magnetic field. Its oscillation period decreases as the interferometer length increases. Right panel: Schematic view of the Mach-Zehnder interferometer. At the beam splitters (dotted curves), mixing between the spin-up channels having opposite valley-isospin occurs. In the

large interferometer, the beam splitters are formed at the intersections of the PN interface with the graphene edge. In the intermediate one, the top beam splitter is formed at the intersection with the edge, while the bottom splitter is at the intersection with the $v = 0$ region in the bulk. In the small one, the two beam splitters are formed in the bulk. **c** $T_{MZ}$ oscillations of the large interferometer plotted at different temperature. The absence of temperature dependence of the average $T_{MZ}$ unambiguously ruled out the effect of the temperature on the intervalley scattering rate.

interferometer. The disappearance is due to the fact that an edge intersection, hence a beam splitter there, cannot be formed below the side gate of $v = 0$, because of spatial separation of the channel of the P region from those of the N region by the $v = 0$ region. The interesting reappearance of the Aharonov-Bohm oscillations, accompanied by the periods much larger than that of the large interferometer, implies the formation of a beam splitter not at the edge intersection but at another position below the side gate. The only possible position, where the channels from the P and N regions can collide, is the "bulk" intersection of the PN interface with the $v = 0$ region inside the graphene bulk (see the schematic views in Fig. 1b). The resulting interferometers are smaller than the large one; in the "intermediate" interferometer, $v_1 = -1$ and $v_2 = 0$, while $v_{i=1,2} = 0$ in the "small" interferometer. Their interferometer length is estimated as $L = 1.05\,\mu m$ and $0.62\,\mu m$ from the location of the bulk intersections in the sample geometry (see Supplementary Fig. S2 in the Supplementary Material). The ratios of the estimated lengths of the three interferometers match with those of their Aharonov-Bohm periods; small mismatch can be interpreted that the spacing between the two interferometer arms (the two spin-up interface channels) slightly differs between the interferometers, depending on the filling factor of the side-gate regions. So it is concluded that a beam splitter is formed inside the bulk rather than at the

edge when $v_{i=1,2} = 0$. The origin of the beam splitter (namely, the valley mixing) at the bulk intersections may be atomic defects, as reported in recent STM experiments[17] on the same source of graphene as ours (NGS graphenium flakes), or many-body states at $v = 0$.

The arms of the three interferometers are expected to be similar. The gate configurations of the three interferometers are the same in the region between the side gates, and different below the side gates. A recent calculation[18] studying edge-channel reconstruction along a PN interface, based on the Chklovski-Shklovskii-Glazman model, estimates that for the gate configuration of our experiment, the spacing between the two arms is 102 nm / 90 nm in the presence/absence of the side gates. Note that this estimation is comparable with 110 nm/ 83 nm obtained from the experimental Aharonov-Bohm period of the large/small interferometer[2]. Therefore, the side gates change the properties of the arm not largely, and it is reasonable to compare the interferometers based on their arm length difference as the first approximation.

We study the interference visibility Vis=$(T_{MZ,max} - T_{MZ,min})/(T_{MZ,max} + T_{MZ,min})$, where $T_{MZ,max\ (min)}$ is the maximum (minimum) value of the oscillation of $T_{MZ}$ (see Fig. 1c). We first discuss thermal decoherence. Figure 2a shows the interference visibility normalized by $V_0$, that is the visibility at base temperature, as a function of

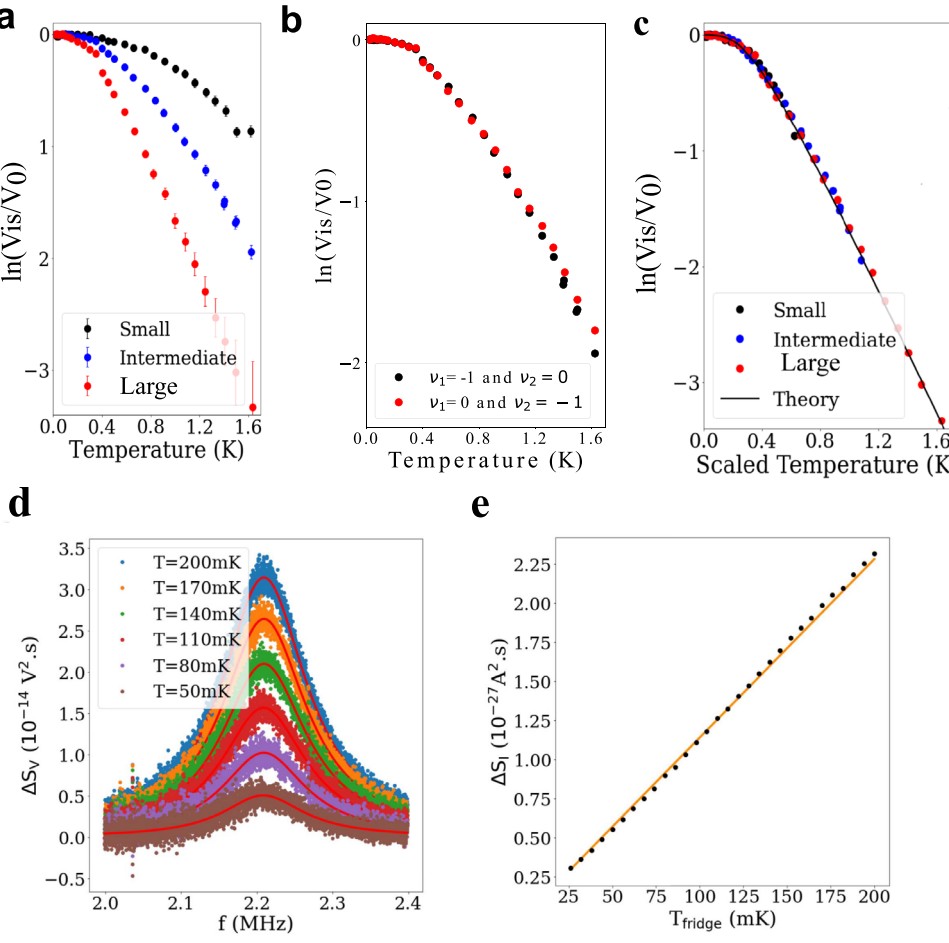

**Fig. 2 | Scaling behavior of thermal decoherence. a** Thermal decay of the interference visibility Vis in log scale for the three interferometers. $V_0$ is the visibility measured at the fridge base temperature. The visibility is defined as: Vis=$(T_{MZ,max}$-$T_{MZ,min})/(T_{MZ,max}+ T_{MZ,min})=T_d/T_s$ with $T_d = T_{MZ,max}$- $T_{MZ,min}$ and $T_s = T_{MZ,max}$+ $T_{MZ,min}$. $T_d$ and $T_s$ and their associated errors $\delta T_d$ and $\delta T_s$ are obtained from the fit. Visibility's error bar is given by $\delta Vis=(T_s*\delta T_d - T_d*\delta T_s)/ (T_s)^2$. **b** Thermal decay of the interference visibility Vis in log scale for the two intermediate interferometer configurations. **c** The decay is redrawn with scaled temperature $LT/L_0$ where $T$ is temperature, $L$ is the interferometer length, and $L_0$ is the length of the large

interferometer. The decay of the three interferometer lies on the same curve, which is in good agreement with an *intra-channel* interaction model. In the theory, we choose the parameters of $v = 4.4 \times 10^4\,m/s$ and $g = 3.3$. **d** Voltage spectral density as a function of the frequency for different refrigerator temperature. The red curves are the theoretical fits given by the Johnson-Nyquist noise of the circuit which is composed of the sample resistance in parallel with an RLC resonator. The gain of the amplification chain is extracted from the fit. **e** Average current noise as a function of temperature.

temperature. Notably, the interference persists above 1.5 K, a temperature much higher than the usual operating temperature of the GaAs edge channels. Below 1 K, the visibility decays not exponentially, but algebraically, which means that thermal decoherence is suppressed. The crossover temperature from the algebraic to exponential regime becomes higher for the smaller interferometers. We have also measured the interference visibility as a function of the temperature for the two intermediate configurations ($v_1 = -1$, $v_2 = 0$ and $\Delta B = 34.5$ mT or $v_1 = 0$, $v_2 = -1$ and $\Delta B = 39$ mT) and observe the same behavior (see in Fig. 2b). The graphene interface channels are quite robust against thermal decoherence even around 1 K and the coherence length, extracted from the exponential decay regime, is 1.24 μm at 1 K.

We note that the algebraic decay does not originate from heating effects. Heating effects are excluded in our case by measurement of electron temperature through the Johnson-Nyquist noise (see the "noise setup" section in the Supplementary Material). We use homemade cryogenic amplifiers combined to a LC tank circuit at 2.2 MHz to avoid the 1/f noise generated by the amplifiers' HEMT and the parasitic noise induced by the dry dilution refrigerator vibrations. After a second stage of amplification at room temperature, the voltage fluctuations are digitized with an acquisition card and the noise spectral density is computed for different refrigerator temperatures (see Fig. 2d). The current fluctuations of the Hall resistance are given by $S_I = \frac{4k_B T_e}{R_H}$ with $k_B$ the Boltzmann constant, $T_e$ the electronic temperature and $R_H = \frac{h}{2e^2}$. The average value of $S_I$ is linear down to a refrigerator temperature of 25 mK (Fig. 2e), confirming that electrons are perfectly thermalized.

Remarkably, the visibility curves lie on a single curve when plotted with temperature scaled by the interferometer length $L$ with a clear crossover from an algebraic to an exponential decay of the visibility that has never been observed in conventional semiconductors (see Fig. 2c). Namely, the visibility does not depend on temperature and the length independently, but only on the product of the two. The crossover temperature, which is inversely proportional to $L$, is 350 mK in the large interferometer. Note that the scaling behavior is satisfied over the large length variation by 300%. This confirms the formation of the interferometers and the beam splitters at the bulk intersections when $v_{i=1,2} = 0$, and also validates that the algebraic decay does not originate from electron heating. In the following we discuss the different decoherence mechanisms that lead to such a scaling behavior.

The algebraic decay implies suppression of thermal decoherence. The algebraic decay has not been reported in GaAs interferometers;

there has been a report on a non-exponential decay that may originate from heating effects[14]. By contrast, in our graphene interferometers being ten times smaller than the GaAs ones[13,14], the universal crossover is clearly observed. The scaling behavior requires a decoherence mechanism to follow the scaling with the interferometer length or have a length scale much longer or shorter than the interferometer length. This excludes disorders or small charge puddles of the bulk from the mechanism. As possible mechanisms, one can cite *inter-channel* interactions between adjacent interface channels and *intra-channel* interactions. The short-range inter-channel interactions can cause decoherence through fractionalization of electron flow into slow and fast modes as in the GaAs edge channels[7,8]. We will show in the following that this latter mechanism is negligible and that the inter-channel interactions are dominated by the intra-channel interactions.

To compare an *intra-channel* interaction model with our experimental observations, we consider a simple capacitive Hamiltonian[5],

$$\mathcal{H}_{\text{int}} = E_C \sum_{\alpha=l,r} \left( \hat{Q}_\alpha/e - N_g \right)^2, \tag{1}$$

where $\hat{Q}_{\alpha=l,r}$ is the charge inside the left and right interferometer arms respectively, $N_g$ is a reference charge number determined by the gate voltages, $E_C = gv\hbar/(2L)$ is the charging energy, $v$ is the drift velocity, and $g$ is a dimensionless interaction parameter. Physically, when an electron enters an interferometer arm, charge density fluctuations in the arm provide which-path information through the capacitive interaction, reducing the interference. The interference visibility, computed with the intra-channel interaction model, satisfies the scaling behavior (Methods), exhibits a universal crossover from the algebraic to the exponential regime, and fits very well the experimental thermal decay (Fig. 2c). The crossover happens at the temperature $T \sim \hbar v/(k_B L)$ comparable with the single-particle level spacing. In the algebraic regime below the crossover temperature, the electron thermal length is longer than the interferometer arms, hence the thermal charge fluctuations and the resulting decoherence by the interaction are suppressed.

By contrast, inter-channel interactions are negligible. We observe the evolution of the visibility with the temperature as we add additional PN interface channels by changing their respective filling factors. In Fig. 3, the visibility decay for $(v_n, v_p) = (4, -1)$ is more or less similar to the $(2, -1)$ case, and the decay for $(2, -2)$ is stronger only slightly. The overall

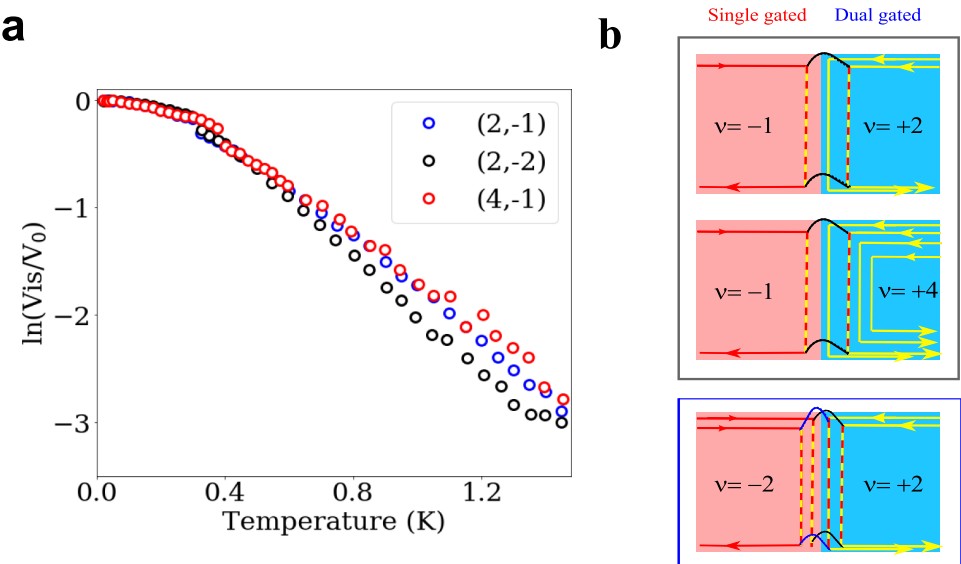

**Fig. 3 | Thermal decoherence at various filling factors. a** Interference visibility Vis versus temperature, as in Fig. 2a, for the large interferometer at $(v_n, v_p) = (2, -1)$, $(2, -2)$, $(4, -1)$. **b** Schematic representation of the different configurations.

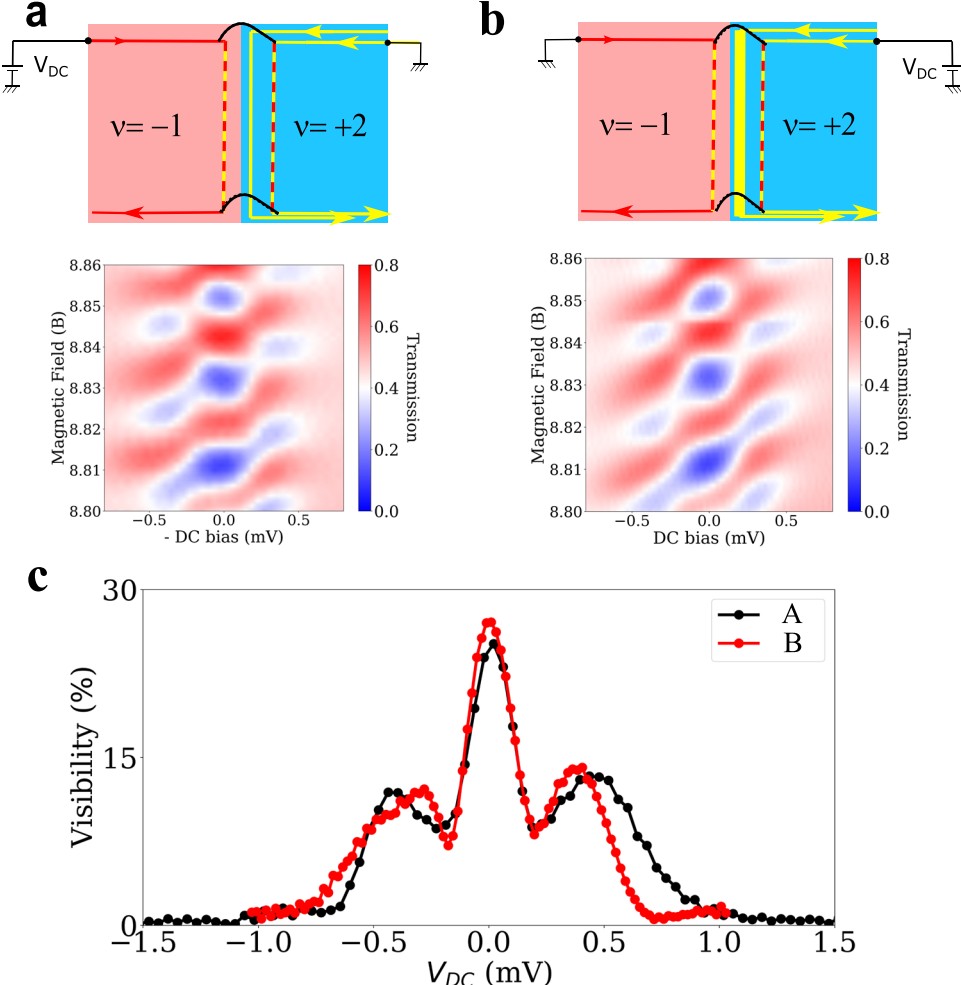

**Fig. 4 | Dependence of lobe pattern on the beam source.** Transmission $T_{MZ}$ of the large interferometer at $(\nu_n, \nu_p) = (2, -1)$ as a function of the magnetic field and the DC bias $V_{DC}$ applied to **a** the upper right ohmic contact or **b** the upper left contact. In **a** (resp. (**b**)), electron beam is injected from the N region of $\nu_n = 2$ (resp. P region of $\nu_p = -1$) to the interferometer, biasing the two (resp. single) edge channels of the region. Note that the DC bias $V_{DC}$ is inverted in **b** for comparison. **c** Interference visibility as a function of $V_{DC}$ in the cases **a**, **b**. It is measured at 9 T. The similarity of the lobe pattern between **a**, **b** implies that inter-edge interactions between the interface channels are weak.

similarities among the decay curves indicate that the inter-channel interactions are not the dominant source of the dephasing, although they may not be completely suppressed. This observation is supported by the geometry of the sample and the configuration of the edge channels. First, in the sample geometry, the vertical distance (30 – 50 nm) between the gates and the graphene layer is shorter than the spacing (50 – 60 nm) between two adjacent edge channels leading to screening of the inter-channel interactions; the spacing is indicated, with assuming a symmetric PN junction, by the 110 nm spacing between the two arms estimated[2] from the Aharonov-Bohm period. This is in sharp contrast with the GaAs edge channels, where the distance between the edge channels and the top gates is typically 90 – 100 nm. Note that in our geometry, the N region has more screening of the inter-channel interactions than the P region, since it is more affected by the top gate. Second, in the $(\nu_n, \nu_p) = (2, -1)$ case, the additional channel is sandwiched between the interferometer arms along the PN interface so that its interaction with the left arm will be similar to its interaction with the right arm if the interactions are present. When it interacts with the two arms equally, our theoretical calculation (see the "theoretical models" section in the Supplementary Material) shows that no decoherence is induced by the interaction. Instead, to fit the thermal visibility decay in Fig. 2 by the calculation based on the inter-channel

interaction, it is required that one arm interacts with the additional channel about 10 times more strongly than the other arm. This asymmetry is unlikely in our setup. In the $(2, -2)$ case, on the other hand, another additional channel is formed outside of the interferometer in the P region, resulting in asymmetry in its interaction with the two arms. In this case, the inter-channel interaction can cause, yet weak, decoherence. In the $(4, -1)$ case, the two more channels are added in the N region in comparison with the $(2, -1)$ case, and they are far apart from the interferometer arms, so decoherence by them will be negligible. These suggest that the inter-channel interactions do not provide the dominant decoherence mechanism.

Next, we examine non-equilibrium decoherence by a finite bias voltage applied to the large interferometer at $(\nu_n, \nu_p) = (2, -1)$. Figure 4 shows the voltage dependence of the visibility, which is called the lobe pattern[14,19–21]. In the pattern, a dip occurs near 220 µV and a single side lobe appears at larger voltages. Interestingly, Fig. 4 shows that the pattern depends only weakly on the beam source, i.e., whether the bias voltage is applied to the upper right ohmic contact (biasing the two edge channels of the N region) or the upper left contact (biasing the single edge channel of the P region). This feature is in contrast with the GaAs interferometers at $\nu = 2$, whose lobe pattern depends largely on whether a bias voltage is applied to only one[14,19,20] or the two edge

channels[21] because of *inter-channel* interactions. Note that our intra-channel interactions model qualitatively reproduces the lobe pattern (see the "theoretical models" section in the Supplementary Material). This confirms that contrary to conventional GaAs interferometers, decoherence in graphene in the quantum Hall regime effect is dominated by *intra-channel* interactions.

Before concluding, we mention recent related works on graphene Fabry-Pérot interferometers[22,23]. In[22] the thermal decay of the interference visibility of the Fabry-Pérot interferometer is discussed for different lengths of the interferometer. The decay is only exponential; no algebraic decay is found probably due to the presence of etch defined edges[24]. However, in a gate defined interferometer[23], no saturation of the visibility at low temperature is observed. The extracted coherence length of 8.1 μm at 32mk is 4.78 times smaller than in the present MZI. A smaller coherence length associated with a larger area may explain the absence of visibility saturation at low temperature in a graphene Fabry-Pérot.

## Discussion

Quantum Hall systems provide a promising platform for the implementation of quantum information processing. Indeed, the one-dimensional dissipationless quantum Hall *edge channels* form an ideal and tunable propagation medium for quantum coherent single-electron wave-packets, the spatial trajectories of which encode the information to process. The basic building blocks of this so-called *flying qubit approach* were demonstrated in GaAs/AlGaAs heterostructures. However, the limited phase coherence length in this material (~20 μm at 20 mK) critically hampers the development of complex multi-qubit architectures needed for quantum information processing. To circumvent this major issue, a new paradigm in terms of material is necessary.

In this work we perform a detailed study of the decoherence processes in a graphene Mach-Zehnder interferometer. While we observe the usual exponential thermal decay of the visibility at high temperature, below a crossover temperature (~350 mK) the decoherence is suppressed. We expect that the presence of the top and bottom gates close to the graphene layer screens interactions between co-propagating edge states, a large source of decoherence. We finally reach a regime where intra channel interactions are the main source of decoherence. The possibility to work in a regime where decoherence is suppressed makes graphene a very promising platform for applications to flying qubits[25–27], orbital entanglement generation[28,29] and valleytronics[30].

## Methods
### Visibility

The visibility noted "Vis" is defined as: Vis = (T$_{MZ,max}$- T$_{MZ,min}$)/(T$_{MZ,max}$+ T$_{MZ,min}$) with T$_{MZ,max}$ the maximum value of the MZI transmission and T$_{MZ,min}$ the min value.

The normalized visibility is defined as Vis/ V$_0$ with V$_0$ the visibility at base temperature.

In ref. 2, a precise study of the transmission (T) dependence of V$_0$ is done that clearly demonstrates the $\sqrt{T(1-T)}$ dependence of the visibility. Nevertheless, it is also shown that the maximum visibility is 60%. It remains unclear why we could not reach 100%.

We note I$_0$ the injected current. The definition of the transmission T$_{MZ}$ depends on the injected current and the pn junction configuration.

In the $v_L = −1 / v_R = +2$ configuration, there is a spectator edge state that does not participate to the interference. The interfering edge states carry the same spin while the spectator edge state carries an opposite spin. The energy cost to flip the spin at 9 T is given by the Zeeman energy g$_\mu$B = 1 mV that is much larger than the electronic temperature: T = 25 mK corresponds to V = k$_B$T/e = 1.2μV. Therefore, the spectator edge channel is fully reflected and half of the current cannot be transmitted leading to a definition of the MZI transmission T$_{MZ}$ = I$_T$/(I$_0$/

2). In the $v_L = −1 / v_R = +4$ configuration, three edge channels from the right region will be reflected and

T$_{MZ}$ = I$_T$/(I$_0$/4). Note that for the voltage-dependent lobe pattern, the visibility is also defined with differential conductance.

### Theoretical model and scaling behavior

We explain the theoretical model based on the intra-edge capacitive interaction model, and show how the scaling behavior appears in the model. The full Hamiltonian is described by

$$\mathscr{H} = \mathscr{H}_0 + \mathscr{H}_{int} + \mathscr{H}_T$$

where $\mathscr{H}_0 = -\hbar v \sum_{\alpha = l,r} \int dx \psi_\alpha^\dagger(x) i \partial_x \psi_\alpha(x)$ is the Hamiltonian for electrons in the left ($\alpha = l$) and right ($\alpha = r$) arms of the MZI. The Hamiltonian $\mathscr{H}_{int}$ for the intra-channel interaction is given in Eq. (1). The Hamiltonian $\mathscr{H}_T = T_U + T_D + h.c.$ describes the MZI beam splitter. Here, $T_U(t) = \hbar v t_U e^{\frac{ieV_{DC}t}{\hbar}} e^{\frac{i\phi_{AB}}{2}} \psi_r^\dagger(0,t)\psi_l(0,t)$ and $T_D(t) = \hbar v t_D e^{\frac{ieV_{DC}t}{\hbar}} e^{\frac{i\phi_{AB}}{2}} \psi_r^\dagger(L,t)\psi_l(L,t)$ describe electron tunneling from the right arm to the left arm at the first and second beam splitters, respectively. $\phi_{AB}$ is the AB phase enclosed by the MZI loop.

To show the universal thermal crossover of the MZI visibility, we consider the regime of small tunneling amplitudes $t_U$ and $t_D$. The current through the MZI is computed,

$$I_T = (|t_D|^2 + |t_U|^2) \frac{e^2}{h} V_{DC} - \left( \frac{e}{\hbar^2} \int dt \left\langle \left[ T_U(0), T_D^\dagger(t) \right] \right\rangle + c.c. \right).$$

The maximum and minimum values of the MZI transmission follows T$_{MZ,max}$ + T$_{MZ,min}$ = 2(|t$_D$|² + |t$_U$|²) and T$_{MZ,max}$ − T$_{MZ,min}$ = $\frac{4\pi}{\hbar^2} |\int dt\, it \langle [T_U(0), T_D^\dagger(t)]\rangle|$. Hence, the visibility of the differential conductance through the MZI at the zero bias limit is given by

$$\text{Vis} = \frac{2|t_D| \cdot |t_U|}{|t_D|^2 + |t_U|^2} |2\pi v^2 \sum_{\eta = \pm} \eta \int dt\, it G^\eta(t)|. \tag{2}$$

$G^\eta(t)$ is an electronic Green's function at finite temperature under the Hamiltonian $\mathscr{H}_0 + \mathscr{H}_{int}$:
$G^+(t) = \langle \psi_l^\dagger(L,t) \psi_l(0,0) \rangle \langle \psi_r(L,t) \psi_r^\dagger(0,0) \rangle$ and $G^-(t) = \langle \psi_l(0,0) \psi_l^\dagger(L,t) \rangle \langle \psi_r^\dagger(0,0) \psi_r(L,t) \rangle$. It is calculated by using the bosonization method,

$$G^\eta(t) = \frac{\exp(\delta\mathscr{G}^\eta(t))}{\left( \frac{2v}{k_B T} \sin[\frac{\pi k_B T}{v}(a - i\eta(L - vt))] \right)^2},$$

$$\delta\mathscr{G}^\eta(t) = i2\eta L \int dq \frac{\frac{g}{2\pi} \left( \frac{\sin qL/2}{qL/2} \right)^2}{1 + \frac{g}{2\pi} \frac{\sin qL/2}{qL/2} e^{-i\eta qL/2}} \frac{e^{i\eta q(L - vt)}}{1 - e^{-qv/k_B T}}$$

where $a$ (> 0) is an infinitesimal length cutoff. $\delta\mathscr{G}^\eta(t)$ comes from the intra-channel interaction.

The scaling behavior Vis(L, T) = Vis(LT) of the visibility can be seen by the fact that Eq. (2) is invariant under the rescaling of variables with a scaling parameter $b > 0$: arm length $L \to bL$, temperature $T \to b^{-1}T$, length cutoff $a \to ba$, momentum $q \to b^{-1}q$, and time $t \to bt$.

Supplementary Materials include the dependence of the thermal decay of the visibility on the interaction parameter $g$, and also the calculation of the dependence of the visibility on the bias voltage for arbitrary tunneling amplitudes $t_U$ and $t_D$.

### Measurements

We used a Cryoconcept dry dilution refrigerator with a base temperature of 13 mK. Measurements of transmitted currents and R$_{Hall}$ values were performed using multiple Lock-in amplifiers with low noise preamplifiers. AC excitations 1nA-5nA with different frequencies

(70Hz-300Hz) were used. Buried ohmic contacts underneath top gates enabled us the direct determination of filling factors from regions of interest.

## Data availability

All data, code, and materials used in the analysis are available in some form to any researcher for purposes of reproducing or extending the analysis.

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

## Acknowledgements

We warmly thank P. Jacques for technical support. Funding: This work was funded by the ERC starting grant COHEGRAPH, the CEA, the French Renatech program, "Investissements d'Avenir" LabEx PALM (ANR-10-LABX-0039-PALM) (Project ZerHall), and by the EMPIR project SEQUOIA 17FUN04 co-financed by the participating states and the EU's Horizon 2020 program. It is also supported by Korea NRF via the SRC Center for Quantum Coherence in Condensed Matter (Gran-tNo.2016R1A5A1008184) and NRF-2019-Global Ph.D. fellowship.

## Author contributions

M.J., P.B., A.A. & P.Rou performed the experiment with help from F.D.P. and P.R.; M.J., P.B., A.A., F.D.P, H.S.S & P.Rou analysed and discussed the data with help from P.R.; J.Y.L, H.S.S, developed the theoretical model and discussed it with P. Rou; T.T., K.W. provided the BN layers; M.J. fabricated the device with inputs from W.D., P.B., A.A., F.D.P & P.Rou; J.Y.L, H.S.S and P.Rou wrote the manuscript with inputs from all coauthors; H.S.S and P.Rou designed and supervised the project.

## Competing interests

The authors declare no competing interests.
