## [Peer Review File · Nature Communications]

REVIEWER COMMENTS

Reviewer #1 (Remarks to the Author):

The authors report experiments on electronic transport in a single-layer graphene pn-junction device in the quantum Hall regime. They observe interference of edge-channels co-propagating along the pn-junction when changing the applied magnetic field. This interferometer is of the Mach-Zehnder type, and the interference is brought about by changing the Aharonov-Bohm phase in the interference loop. Two additional top gates above the p-region allow them to tune the co-propagation length and hence the relevant length of the interfering beams.

It seems to me that the new data provided in the paper (as compared to their recent PRL, Ref. 2 in the paper) is quite limited. The phenomena shown in Fig. 1 and 4 of the presented manuscript have been reported by them before in Ref. 2. New are the data in Figs 2 and 3.

It remains unclear, if the manuscript is based on data taken on the same device as in Ref. 2.

The focus of the paper is on the decoherence mechanism limiting the visibility of the interference. The observation in Fig. 2 is, that decoherence at high temperatures is exponentially suppressed with increasing temperature, whereas at sufficiently low temperatures, the exponential visibility increase saturates. They find that the observed dependence of the visibility on temperature scales with the effective length of the interferometer. They try to get an experimental handle on the importance of inter-channel decoherence by tuning the number of edge channels in the n- and p-regions, as shown in Fig. 3, and find only a very weak dependence of the temperature-dependent visibility on the number of edge channels. Applying a model for intra-channel decoherence, they find good agreement with their scaled data.

While the scaling of the temperature dependence of the visibility is an interesting result, the manuscript has in my opinion several weaknesses. First, the authors claim that the saturation of the visibility increase at the lowest temperatures is not due to heating, and mention noise measurements in the supplemental material. This data should certainly be presented in the main paper, as it is a strong piece of evidence. Second, the exclusion of inter-channel decoherence, while plausible, is not entirely convincing based on the experimental data. Third, the model used for fitting the data is not sufficiently described in the main text. In particular, the scaling-behavior of the model is not explicitly presented.

Details of this criticism and further minor suggestions are given below. Based on my assessment of the paper, I do not recommend publication of the manuscript in Nature Communications in its present form.

More detailed comments and questions to be addressed in the manuscript:

1. What is the thickness of the hBN encapsulating layers?
2. Is the orientation of the graphene lattice with respect to the pn-junction known? Is it relevant?
3. Is this the same device as the one used in Ref. 2?
4. What is the electronic temperature for the measurements presented in Fig. 1?
5. In Fig. 1: what is the absolute magnetic field for this measurement?
6. In Fig. 1B: indicate the period of the oscillations with arrows labeled by the numbers quoted in the main text. It seems to me that the period of the large interferometer is 20 mT rather than the 25 mT given in the manuscript.
7. In Fig. 1b: $\nu_1 = -1$, $\nu_2 = 0$; can they show data with $\nu_1 = 0$, $\nu_2 = -1$? Does it give the same period?
8. In Fig. 2B: should the horizontal axis not be $T \times L$ in units of $k_B \mu_m$? The same visibility is reached in the shorter interferometer, if the temperature is increased, i.e., if $T \times L$ stays constant. Does this not mean that $\ln(\text{Vis}/V_0) \propto T \times L$, meaning that plotting $\ln(\text{Vis}/V_0)$ vs. $T \times L$ collapses the curves on top of each other? This would also be in agreement with thermal fluctuations in the higher temperature regime, for which $\ln(\text{Vis}/V_0) \propto -L/L_\phi$ and $L_\phi \propto 1/T$.
9. Is the time scale an electron spends in the interferometer relevant for the decoherence? Which frequency range of decohering fluctuations is this experiment sensitive to? Could decoherence also be caused by charge noise in the graphene or in the boron nitride, i.e., by $1/f$ noise?
10. The authors find that adding channels to the n-region does not cause stronger decoherence. Based on this observation they argue that inter-channel decoherence is negligible in their experiment. However, it seems to me that the observation only shows that the inter-channel decoherence is only negligible for the added edge channels, which are spatially further separated, but not necessarily for the one present at $\nu=2$.
11. In the same context, the authors compare the separation of the two interferometer arms (110 nm), with the hBN thickness (50-60 nm) separating the gate from the channels, if I understand correctly (the wording in this paragraph is not clear). They then argue that the proximity of the gate would lead to a significant screening of inter-edge decoherence. There is, however, no experimental evidence for that in the data presented. Do the authors have data from experiments on samples with different hBN thicknesses?

Reviewer #2 (Remarks to the Author):

The present manuscript reports on the temperature dependence of interference visibility in a graphene-based electronic Mach-Zender interferometer. The interferometer is formed along a gate-defined pn junction. The length of the interferometer can be varied using two additional gated regions on both sides of the pn interface, and three different lengths are studied. Their respective boundary conditions change depending on whether the edge channels mixing occurs at the sample edge or in the bulk, as both situations strongly differ in terms of intervalley scattering required to observe the interferences. The operating mechanism of the interferometer has been extensively discussed in Ref.2, and though its recall is rather rude for a broad audience journal such as Nature Communications, it might be enough to understand roughly for a non-expert solid-state physicist. The temperature dependence of the observed interference visibility is studied, and the authors evidence a scaling behavior of the visibility, shown to scale algebraically at low temperature, and exponentially at higher temperature. When rescaled by the interferometer length, the curves of visibility versus temperature are found to collapse on the same curve. A possible saturation of the electronic temperature is convincingly discarded. The scaling behavior is then analyzed in different filling factor configurations, and the visibility as a function of source-drain and drain-source biases is analyzed. These two experimental analyses are claimed to support a model pointing towards intra-channel interactions as the main decoherence source, and discarding inter-channel ones.

While we find convincingly reported that the temperature dependence of the interference fringes saturates, and well argued that it sounds remarkable, we find that the explanation in terms of intra-channel interactions fall quite short of convincing evidences.

Overall, we find the paper very technical and not self-contained, since many simple experimental details are not provided and required the cautious reading of Ref. 2 for us to understand. The authors discuss extensively the tuning of the interferometer size, which is easy to understand and could be strongly summarized, while the message clarity would gain from more details on the central results, i.e. the unusual temperature dependence of the visibility. Written as it is, the manuscript sounds like an extension of Ref. 2, in which equivalents of Figures 1 and 4 of the present manuscript were already shown. Hence this manuscript would perfectly fit a specialized journal like Physical Review B. We do not recommend its publication in Nature Communications in its actual state, unless major revisions are made.

However, we are convinced of the results novelty, and that the present findings may be of great importance for future developments of electron optics experiments in graphene quantum Hall edge channels. Therefore, a more pedagogical, self-contained and better argued version of the manuscript might meet the standards of Nature Communications. Below are some questions and hints that could help making the paper understandable for a broader audience.

1) All the paper's figure present normalized visibility, which is never defined properly. It is very hard to figure out what is "Vis", and when trying to guess its definition from the oscillations reported in Figure 1, it appears clear that it shall be associated with an error bar. As an example in Fig.1b in the intermediate case the amplitude of the oscillations vary by 50% from one to another. Figure 1 would largely benefit from oscillations plotted at different temperature, and a visual explanation of what is considered as the visibility.

2) The transmission itself is unclearly defined. It requires to go in the supplementary to figure out that it is in fact $IT/(I0/2)$, and to read Ref. 2 to understand why it is normalized by $(I0/2)$. Once understood that it is because one of the two channels cannot presumably scatter to the other side because spin-flips are forbidden energetically (which could be worth recalling and justifying for non-expert readers by the way), it strikingly appears that this definition shall change when considering different filling factors as is done in Fig.3. Could the authors explain the used definitions and the observed amplitude in the different configurations? It shall not change the results by any means since it is normalized to $V0$ but it would clarify the measured quantities for the readers. Furthermore, the "Methods" section should be widely improved and include at least the definitions of Vis, $V0$ and TMZ.

3) Two middle-size interferometers are possible given the sample geometry (with a beam splitter at the first or the second side gate), why did the authors chose to discuss only one? Valley scatterers must exist in the bulk on both sides since the authors can form the small interferometer. The "universality" of the scaling shown in Fig.2b would benefit from a fourth example with same length as the presented intermediate.

4) Did the authors also check the opposite polarity (i.e. holes in the left part and electrons on the right, like (2,-1) configuration?). And same question for the (-2,1) configuration? If not could the authors just comment on what prevented these explorations in the experiment? Given the different distances from the gates the claims may benefit from these additional data if they support the authors interpretation.

5) Line 106-107, it is argued that the 3 interferometers "share the same PN interface, hence, the same properties of electron velocity and interaction strengths." But the additional length of the intermediate and large interferometers have different gate configurations due to the presence of the side gates, closer to the edge channels than the top gate. According to the authors themselves, the distance of the metallic gate to the channels is of crucial importance in the mitigation of interactions through screening. We then respectfully disagree with this sentence, and believe that the "length" of the different interferometers shall not be the only thing to be taken into account.

6) In the same vein, the comparison between the inter-channel distances and their distances to the gates is very confusing (lines 154-158 where it is not clear if it is the distance in the 2DEG plane or the vertical distance that is considered), so is the comparison with GaAs Mach-Zender interferometers. The reported distance variations are not very convincing (differing by roughly a factor 2-3), and may not be enough for a direct comparison: wouldn't the different dielectric constant of GaAs vs h-BN also play a noteworthy role in this discussion?

7) In the main text, it is said that "Below 1 K, the visibility decays not exponentially, but algebraically, which means that thermal decoherence is suppressed". The link between the algebraic decay and the suppressed decoherence should be further justified. The article, in its present form, does not allow to establish this link properly (maybe this is just the word "suppressed" that should be replaced by "weak"). Furthermore, the algebraic and exponential decays are presented as two distinct regimes separated by a crossover temperature. However, these two behaviors are captured by the same model detailed in the supplementary materials. Could the authors elaborate on that point, and pinpoint the physical meaning of the crossover temperature based on the model output?

8) The model Hamiltonian sounds quite phenomenological, but no insight is given in the text concerning why this form is expected. We find it rather crude for Nature communication. Could the authors give at least a hand-waving flavor for this Hamiltonian? And precisely which gate does V_g refer to at the same time? Is it expected to be the same gate voltage for both sides or shall it be two different V_{gi} ? Besides, the model in the supplementary materials lacks references to be better justified.

9) It is hard to understand, besides the geometrical argument of the gate distances, what is the specificity of graphene in this peculiar thermal behavior? We could draw no clear explanation from the text, which would give a feeling of the reason for this intra-channel dominating. In particular, we don't really understand why the authors compare their results only with GaAs-Mach-Zender interferometers (including a lot of self-citations), while they don't discuss recent results in graphene Fabry-Pérot interferometers (e.g. Déprez Nat. Nano. 2021, Ronen Nat. Nano. 2021). It could strengthen the paper if differences were discussed and if conclusions can be drawn on why the interferometers cited above do not reach this intra-channel dominated decoherence. Is it only a matter of length? If so are such interferometers also expected to reach this regime if designed smaller?

10) Other possible mechanisms for decoherence are not ruled out. In particular, we naively think of additional scattering events between the two arms of the interferometer itself. If intervalley scatterers do exist in the bulk (a pre-requisite need for observing interferences in the intermediate and small interferometers), could they also couple the co-propagating channels across the pn interface? What would then be the effect on the coherence?

11) Could the author discuss at some point in the text what limits V_0 to a non-unitary value? Is it only the coupling coefficients at the ends of the interferometers as in standard MZ interferometry or could the effect pointed in question 10 play a role in this finite visibility? Once again from Ref.2 insights can be obtained on the role of T1 in the visibility but nothing is recalled in the present manuscript, and unless the paper is addressed only to MZ specialists it could be worth recalling it in one or two simple sentences.

12) A naïve question related to question 11: could temperature have a role on the intervalley scattering rate? If not could the authors unambiguously rule this out such that no doubt can remain on this effect contribution to visibility?

13) Concerning the two models of intra and inter-channels interaction, we find the discussion very imbalanced in the favor of the scenario the authors want to demonstrate (namely intra-channel interaction dominating decoherence). For intra-channel model they apply an phenomenological decoherence with source-drain by multiplying the visibility by $\exp(-V^2/V_0^2)$, which makes Fig. S8B looking very much like the experimental data. On the other hand, they apply another formalism to capture the decoherence (ref. 7 in the supplementary) for the second model (inter-channel), which makes its results plotted in Fig. S9B looking very different from the experimental data (if we are not mistaking). Could the authors apply the same formalism to both models before comparing the results?

14) In the supplementary materials, we believe the scales are wrong in Figs. S3C, S4B, S4C, S5B and S5C, and that the authors meant V instead of mV.

15) The conclusion of the paper is very elusive and projects the reader to very speculative applications regarding the paper's core (except maybe valleytronics), while not at all summing up and clarifying the results. A real conclusion on the findings of the paper would be appreciable.

Boris Brun

Reviewer comments:

Reviewer #1 (Remarks to the Author):

The authors report experiments on electronic transport in a single-layer graphene pn-junction device in the quantum Hall regime. They observe interference of edge-channels co-propagating along the pn-junction when changing the applied magnetic field. This interferometer is of the Mach-Zehnder type, and the interference is brought about by changing the Aharonov-Bohm phase in the interference loop. Two additional top gates above the p-region allow them to tune the co-propagation length and hence the relevant length of the interfering beams.

It seems to me that the new data provided in the paper (as compared to their recent PRL, Ref. 2 in the paper) is quite limited. The phenomena shown in Fig. 1 and 4 of the presented manuscript have been reported by them before in Ref. 2. New are the data in Figs 2 and 3.

It remains unclear, if the manuscript is based on data taken on the same device as in Ref. 2.

The focus of the paper is on the decoherence mechanism limiting the visibility of the interference. The observation in Fig. 2 is, that decoherence at high temperatures is exponentially suppressed with increasing temperature, whereas at sufficiently low temperatures, the exponential visibility increase saturates. They find that the observed dependence of the visibility on temperature scales with the effective length of the interferometer. They try to get an experimental handle on the importance of inter-channel decoherence by tuning the number of edge channels in the n- and p-regions, as shown in Fig. 3, and find only a very weak dependence of the temperature-dependent visibility on the number of edge channels. Applying a model for intra-channel decoherence, they find good agreement with their scaled data.

While the scaling of the temperature dependence of the visibility is an interesting result, the manuscript has in my opinion several weaknesses. First, the authors claim that the saturation of the visibility increase at the lowest temperatures is not due to heating, and mention noise measurements in the supplemental material. This data should certainly be presented in the main paper, as it is a strong piece of evidence. Second, the exclusion of inter-channel decoherence, while plausible, is not entirely convincing based on the experimental data. Third, the model used for fitting the data is not sufficiently described in the main text. In particular, the scaling-behavior of the model is not explicitly presented.

We would like to thank reviewer #1 for their constructive comment. The experimental study of the decoherence in a graphene interferometer in the quantum Hall regime is a very recent topic, stemming from the realization of the first tunable Mach-Zehnder interferometer (MZI). In our work, we go beyond the demonstration of the tunability of the MZI (Ref. 2) and show two regimes of decoherence: a regime of negligible decoherence at low temperature whereas at high temperature the visibility is exponentially suppressed. It is shown that the intra-channel interaction is the main decoherence source. The fabrication of such a tunable MZI is extremely demanding and it is true that data have been taken on the same device as in Ref. 2. Before responding to the reviewer's remarks point by point, we would like to emphasize the differences of our paper with Ref. 2:

1. While the demonstration of the large MZI has been done in Ref. 2, the possibility of tuning the PN junction to obtain a medium or small interferometer like the one in Fig. 1 was not mentioned in Ref. 2. These results are new and important because they allow the study of decoherence processes under the same conditions e.g. the same device, the same cooling.

2. Between Ref. 2 and the present manuscript, we performed a profound modification of the experiment with the implementation of a shot noise measurement line to measure the electronic temperature and additional filtering to ensure that the electrons are correctly thermalized. This was a fundamental aspect of our study to ensure that the visibility saturation did not simply come from poor thermalization of the electrons. Also note that while the device is the same, the measurements in Ref. 2 were performed for a different cooling. As pointed by the Reviewer, data in Ref. 2 show a periodicity in magnetic field of 25mT while in the present manuscript it is 20mT. The experimental set up modification together with a new cooldown lead to a slightly different behavior of the sample. We clarify this point in the main text.
3. We would like to point out that the measurements in Ref. 2 and in Fig. 4B were taken at different times (specifically, 6 months separate these two measurements). The measurement in Fig. A also required a modification of the experimental set up since the DC bias is now applied to the left contact. Some filtering to obtain a symmetric configuration with Fig. B was added and carefully characterized.

First, the authors claim that the saturation of the visibility increase at the lowest temperatures is not due to heating, and mention noise measurements in the supplemental material. This data should certainly be presented in the main paper, as it is a strong piece of evidence.

The reviewer is right to mention the importance of this measurement. A trivial explanation when the visibility saturates could come from poor thermalization of the interfering electrons. The only way to rule this out is to measure the Johnson Nyquist noise which is an accurate measurement of the absolute electronic temperature. Following the reviewer's comment, we now explicitly discuss this measurement in the main text, while we have extended discussion on the experimental noise set up and the calibration process in the supplementary material.

Second, the exclusion of inter-channel decoherence, while plausible, is not entirely convincing based on the experimental data

The inter-channel decoherence model has been introduced in Ref. 7 to explain the lobe type structure in the visibility as a function of the applied DC bias in GaAs interferometers. We would like to remind to the reviewer the main experimental results that have been reported in GaAs interferometers:

1. In the initial paper describing the MZI in GaAs [Nature volume 422, pages 415–418 (2003)], the dependence of the visibility of the interference pattern on the applied voltage is shown in Fig. 3a and no lobe type structure is observed. In this experiment, the DC bias is applied equally to both edge states that are injected into the MZI.
2. In a second experiment from the same group [Phys. Rev. Lett. 96, 016804, 2006], an additional quantum point contact is added that allows selective transmission of edge channels toward the MZI. Tuning this transmission to 1, the two edge states injected into the MZI are now out of equilibrium. While the chemical potential of the outer edge channel that interferes is at $\mu = -eV_{DC}$, the inner edge channel is now at zero chemical potential. It leads to lobe type structure in the visibility.
3. In a third experiment [Phys. Rev. B 79, 245324, 2009], the transmission of the first MZI quantum point contact is continuously changed from 1 to 0 and the dependence of the visibility with the DC bias is discussed. In the weak transmission configuration, a lobe type structure is observed. In the transmission close to 1, the behavior of the visibility is qualitatively very different since there is a local minimum of the visibility at zero bias, then the visibility increases until 10 μ V and finally decreases at larger bias (see in Fig. 3C).

The inter-channel decoherence model of Ref. 7 successfully explains the measurements reported in the last three papers. Charge fractionalization induced by the inter-channel interaction is the main source

of the decoherence. Similarly, in ref [6], it is shown that an inter-channel capacitive coupling is the underlying mechanism responsible for the finite temperature coherence time of edge states in GaAs.

We agree that the measurements presented in Figure 2 are not sufficient to distinguish the inter-channel interaction model from the intra-channel interaction model. But the measurements shown in Figure 3 and 4 demonstrate that the inter-channel interactions cannot explain our data.

In Figure 4, applying the DC bias on the $\nu = -1$ region, the lobe type structure of the visibility as reported in GaAs is observed. On the other hand, applying the DC bias on the $\nu = +2$ region should suppress the lobe type structure as observed in the initial paper describing the MZI in GaAs [Nature volume 422, pages 415–418 (2003)]. Since we observe the same visibility pattern, the underlying decoherence mechanism must be different from that of GaAs.

In Figure 3, we discuss the effect of additional edge states on the visibility, a very interesting possibility that cannot be probed in GaAs. Inclusion of additional inner edge channels to the MZI (see in Figure 3B), by changing the filling factors of the PN junction, does almost not affect the thermal dephasing.

Third, the model used for fitting the data is not sufficiently described in the main text. In particular, the scaling-behavior of the model is not explicitly presented.

We discuss the theoretical model in more details both in the main text and Method. The scaling behavior is explicitly mentioned in the main text, and proved in Method.

More detailed comments and questions to be addressed in the manuscript:

1. What is the thickness of the hBN encapsulating layers?

The bottom BN layer is equal to 33nm and the top one is equal to 27nm. We have added a detailed description of the device in the supplementary (see in Fig. S3).

2. Is the orientation of the graphene lattice with respect to the pn-junction known? Is it relevant?

The orientation of the graphene lattice with respect to the pn-junction is not known. On the other hand the microscopic properties of the physical edge is crucial. It has been shown in Ref. 2 that the degree of valley mixing at the intersection between the PN junction interface and the physical edge of the sample is given by the microscopic properties of the physical edge, zigzag, armchair or disordered. The degree of valley mixing gives the transmission of each valley splitter and the amplitude of the current oscillations visibility. So Reviewer 1 is correct in a way to say that the orientation of the graphene lattice with respect to the pn-junction is important. Our theory, which is a microscopic theory, includes it.

3. Is this the same device as the one used in Ref. 2?

As discussed previously, it is the same sample as the one used in Ref. 2 but measured in different conditions (different electrostatic environment, different cool down). Note that we consider another aspect of the device since we probe three different lengths (not mention in Ref. 2).

4. What is the electronic temperature for the measurements presented in Fig. 1?

The Johnson Nyquist noise measurement confirms the electrons are perfectly thermalized to the dilution refrigerator, and the electronic temperature is equal to the base temperature which is 25mK.

5. In Fig. 1: what is the absolute magnetic field for this measurement?

For the large MZI, the absolute magnetic field is 9T, for the medium one 8.4T and the small one 7.8T.

6. In Fig. 1B: indicate the period of the oscillations with arrows labeled by the numbers quoted in the main text. It seems to me that the period of the large interferometer is 20 mT rather than the 25 mT given in the manuscript.

Reviewer is right to note that in Fig. 1B the period of the large interferometer is 20mT. We correct this in the main text.

7. In Fig. 1b: $\nu_1 = -1$, $\nu_2 = 0$; can they show data with $\nu_1 = 0$, $\nu_2 = -1$? Does it give the same period?

When $\nu_1 = -1$ and $\nu_2 = 0$, we measure a period of 34mT. When $\nu_1 = 0$ and $\nu_2 = -1$, we measure a period of 39mT. We have measured the dependence of the visibility with the temperature for the two configurations: $\nu_1 = -1$, $\nu_2 = 0$ and $\nu_1 = 0$, $\nu_2 = -1$. We do not observe any difference. We have added these precisions in the main text and a new figure where the thermal decay of the visibility is shown for the two configurations.

8. In Fig. 2B: should the horizontal axis not be $T \times L$ in units of $k_B T$? The same visibility is reached in the shorter interferometer, if the temperature is increased, i.e., if $T \times L$ stays constant. Does this not mean that $\ln(\text{Vis}/V_0) \propto T \times L$, meaning that plotting $\ln(\text{Vis}/V_0)$ vs. $T \times L$ collapses the curves on top of each other? This would also be in agreement with thermal fluctuations in the higher temperature regime, for which $\ln(\text{Vis}/V_0) \propto -L/L_{\text{phi}}$ and $L_{\text{phi}} \propto 1/T$.

We thank the reviewer for having noticed this typo. The scaled temperature should be $L T / L_0$ and not $L_0 T / L$ as written in caption of the Figure 2. We have corrected the caption.

9. Is the time scale an electron spends in the interferometer relevant for the decoherence? Which frequency range of decohering fluctuations is this experiment sensitive to? Could decoherence also be caused by charge noise in the graphene or in the boron nitride, i.e., by $1/f$ noise?

As the reviewer mentions, the residence time L/v within which an electron spends in the interferometer is relevant. If the residence time is longer, an interfering electron has more chance of interacting with other fluctuating electrons and hence suffers more decoherence. This is why the larger MZI exhibits more suppression of the interference. Note that assuming the drift velocity of $4.4 \times 10^4 \text{m.s}^{-1}$, we estimate the residence time of the large, intermediate, and small interferometers as 34ps, 24ps, and 14ps, respectively. In other words, the frequency range of the decohering fluctuations is set by v/L , the inverse of the residence time or the single particle level spacing of the interferometer arms. The crossover from the algebraic to exponential decay of the visibility happens around the temperature $T \sim \hbar v / (k_B L)$. We revise the manuscript to include this discussion.

On the other hand, the reviewer also mentions the $1/f$ noise. The $1/f$ noise is linked to statistical fluctuations due to some inhomogeneities or defects trapped in the conductor. The defect will absorb or emit electrons. If the defect is close enough to one arm of the interferometer, this can lead to

dephasing or decoherence. It can also change the resistance of the sample either by changing the electronic density or by modifying the electrostatic potential seen by the conducting electrons. The power spectral density associated with a single defect is given by:

$$S_N(\omega) = 4\overline{\Delta N^2} \frac{\tau}{1 + \omega^2\tau^2}$$

Here τ is the typical time scale during which the defect absorbs or emits electrons and ΔN is the fluctuation of the involved charges. This noise is also called telegraphic noise since it involves two-level systems: the electron is trapped or not and the power spectral density has a Lorentzian shape. The $1/f$ is obtained when several defects with different time scales τ_i are involved. If we assume that τ follows a thermally activated law $1/\tau \propto e^{-E/k_B\theta}$ with E the height of the potential barrier and θ the temperature, and if the distribution of E is uniform on an interval $[E_{min}, E_{max}]$, averaging over all the defects gives a power spectral density:

$$S_N(f) = \frac{1}{2\pi f} \frac{k_B\theta}{E_{max} - E_{min}} (\arctan(\omega\tau_{max}) - \arctan(\omega\tau_{min}))$$

If there is $1/f$ noise in our sample, we should detect it by measuring the noise as a function of the temperature. In the main text Figure 2D, we have added the power spectral density measured in our sample for different temperatures. At large temperature (200mK), we do not observe any enhancement of the noise a lower frequency that would be the signature of a $1/f$ noise in our sample.

To further confirm it, we have measured the shot noise of a valley splitter as a function of a given transmission. Fig S10 shows the excess shot noise defined as $\Delta S_I(V_{DC}) = S_I(V_{DC}) - S_I(0)$ as a function of the applied bias V_{DC} for transmission $D = 0.5$. We compare $\Delta S_I(V_{DC})$ to the expected shot noise auto-correlation: $\Delta S_I(V_{DC}) = \frac{2e^2}{h} \sum_n D_n(1 - D_n) \left[eV_{DC} \coth \frac{eV_{DC}}{2k_B T} - 2k_B T \right]$. In the present case $\sum_n D_n(1 - D_n) = 2D(1 - D)$ for the two partitioned modes (two pseudo-spin). The agreement with the expected shot noise confirms the absence of $1/f$ noise that would be induced by defects close to the valley splitter.

We have added this discussion in the Supplementary material.

10. The authors find that adding channels to the n-region does not cause stronger decoherence. Based on this observation they argue that inter-channel decoherence is negligible in their experiment. However, it seems to me that the observation only shows that the inter-channel decoherence is only negligible for the added edge channels, which are spatially further separated, but not necessarily for the one present at $nu=2$.

Reviewer 1 mentions a valid and important point. Figure 3 shows that the visibility decay for $(\nu_n, \nu_p) = (4, -1)$ is more or less similar to the $(2, -1)$ case, and the decay for $(2, -2)$ is stronger slightly. The overall similarities among the decay curves indicate that the inter-channel interactions are not the dominant source of the dephasing, although they may not be completely suppressed. This observation is supported by the geometry of the sample and the configuration of the edge channels. First, in the sample geometry, the distance (30 - 50 nm) between the gates and the graphene layer is shorter than the spacing (50 - 60 nm) between two adjacent edge channels leading to screening of the inter-channel interactions; the spacing is indicated, with assuming a symmetric PN junction, by the 110 nm spacing between the two arms estimated from the Aharonov-Bohm period. This is in sharp contrast with the GaAs edge channels, where the distance between the edge channels and the top gates is typically 90 - 100 nm. We note that in our geometry, the N region has more screening of the inter-channel interactions than the P region, since it is more affected by the

top gate. Second, in the $(\nu_n, \nu_p) = (2, -1)$ case, the additional channel is sandwiched between the interferometer arms along the PN interface so that its interaction with the left arm will be similar to its interaction with the right arm if the interactions are present. When it interacts with the two arms equally, our theoretical calculation (Supplementary Material) shows that no decoherence is induced by the interaction. Instead, to fit the thermal visibility decay in Fig. 2 by the calculation based on the inter-channel interaction, it is required that one arm interacts with the additional channel about 10 times more strongly than the other arm. This asymmetry is unlikely in our setup. In the $(2, -2)$ case, on the other hand, two additional channels are formed between the arms, and one of the channels is located closer to one arm than the other, resulting in asymmetry in its interaction with the two arms. In this case, the inter-channel interaction can cause, yet weak, decoherence. In the $(4, -1)$ case, the additional channels are in the N region, and they are far apart from the interferometer arms, so decoherence by them will be negligible.

While the first point was already discussed in the manuscript, the second point is newly included in the revised version of our manuscript. Supplementary Material is also updated to include the supporting calculation. The above points, combined with the lobe pattern structure (which is independent of whether the bias voltage is applied at the right or left ohmic contact) in Fig. 4, strongly suggest that the inter-channel interactions do not provide the dominant decoherence mechanism.

11. In the same context, the authors compare the separation of the two interferometer arms (110 nm), with the hBN thickness (50-60 nm) separating the gate from the channels, if I understand correctly (the wording in this paragraph is not clear). They then argue that the proximity of the gate would lead to a significant screening of inter-edge decoherence. There is, however, no experimental evidence for that in the data presented. Do the authors have data from experiments on samples with different hBN thicknesses?

The reviewer is correct. For the sake of clarity, we have reworked this part of the manuscript, providing more convincing arguments why the inter-channel interactions cannot be a dominant decoherence mechanism (see our answer to the above comment 10).

Performing a systematic study as a function of the hBN thickness would be interesting but is beyond the scope of this paper. Fabricating a functional and tunable Mach Zehnder interferometer was extremely difficult and required several months of development. Instead, we would like to draw the reviewer's attention to a recent theoretical work [arXiv:2201.12025]. According to this work, the spacing between the PN interface channels is essentially determined by the distance between the graphene to the gates. As the distance becomes smaller, the spacing decreases. On the other hand, more screening is expected when the distance is smaller than the spacing. Therefore, the dependence of the screening on the hBN thickness may not be a simple question. More theoretical works will be useful

Reviewer #2 (Remarks to the Author):

The present manuscript reports on the temperature dependence of interference visibility in a graphene-based electronic Mach-Zehnder interferometer. The interferometer is formed along a gate-defined pn junction. The length of the interferometer can be varied using two additional gated regions on both sides of the pn interface, and three different lengths are studied. Their respective boundary conditions change depending on whether the edge channels mixing occurs at the sample

edge or in the bulk, as both situations strongly differ in terms of intervalley scattering required to observe the interferences. The operating mechanism of the interferometer has been extensively discussed in Ref.2, and though its recall is rather rude for a broad audience journal such as Nature Communications, it might be enough to understand roughly for a non-expert solid-state physicist. The temperature dependence of the observed interference visibility is studied, and the authors evidence a scaling

behavior of the visibility, shown to scale algebraically at low temperature, and exponentially at higher temperature. When rescaled by the interferometer length, the curves of visibility versus temperature are found to collapse on the same curve. A possible saturation of the electronic temperature is convincingly discarded. The scaling behavior is then analyzed in different filling factor configurations, and the visibility as a function of source-drain and drain-source biases is analyzed. These two experimental analyses are claimed to support a model pointing towards intra-channel interactions as the main decoherence source, and discarding inter-channel ones.

While we find convincingly reported that the temperature dependence of the interference fringes saturates, and well argued that it sounds remarkable, we find that the explanation in terms of intra-channel interactions fall quite short of convincing evidences.

Overall, we find the paper very technical and not self-contained, since many simple experimental details are not provided and required the cautious reading of Ref. 2 for us to understand. The authors discuss extensively the tuning of the interferometer size, which is easy to understand and could be strongly summarized, while the message clarity would gain from more details on the central results, i.e. the unusual temperature dependence of the visibility. Written as it is, the manuscript sounds like an extension of Ref. 2, in which equivalents of Figures 1 and 4 of the present manuscript were already shown. Hence this manuscript would perfectly fit a specialized journal like Physical Review B. We do not recommend its publication in Nature Communications in its actual state, unless major revisions are made.

However, we are convinced of the results novelty, and that the present findings may be of great importance for future developments of electron optics experiments in graphene quantum Hall edge channels. Therefore, a more pedagogical, self-contained and better argued version of the manuscript might meet the standards of Nature Communications.

We would like to thank the Reviewers for their positive comments and to give us the possibility to improve our manuscript.

Below are some questions and hints that could help making the paper understandable for a broader audience.

1) All the paper's figure present normalized visibility, which is never defined properly. It is very hard to figure out what is "Vis", and when trying to guess its definition from the oscillations reported in Figure 1, it appears clear that it shall be associated with an error bar. As an example in Fig.1b in the intermediate case the amplitude of the oscillations vary by 50% from one to another. Figure 1 would largely benefit from oscillations plotted at different temperature, and a visual explanation of what is considered as the visibility.

To avoid any confusion, we have followed the Yacoby's group paper definition of the visibility noted "Vis" that is defined as : $Vis = (T_{MZ,max} - T_{MZ,min}) / (T_{MZ,max} + T_{MZ,min})$ with $T_{MZ,max}$ the maximum value of the MZI transmission and $T_{MZ,min}$ the min value defined by measuring the differential conductance. We add this definition in the main text of the manuscript which is also shown in Figure 1C. We completely agree with the Reviewer that it shall be associated with error bars which are shown in Figure2A for all the interferometer sizes (the definition is given in the supplementary). To extract the

visibility, we ensure a reasonable fit with a sine curve over two or three periods and we change the temperature. We will add oscillations for different temperature in Figure 1C with a visual explanation of the visibility. For the sake of clarity, we have largely reworked the beginning of the manuscript, defining properly what is measured.

2) The transmission itself is unclearly defined. It requires to go in the supplementary to figure out that it is in fact $I_T/(I_0/2)$, and to read Ref. 2 to understand why it is normalized by $(I_0/2)$. Once understood that it is because one of the two channels cannot presumably scatter to the other side because spin-flips are forbidden energetically (which could be worth recalling and justifying for non-expert readers by the way), it strikingly appears that this definition shall change when considering different filling factors as is done in Fig.3. Could the authors explain the used definitions and the observed amplitude in the different configurations? It shall not change the results by any means since it is normalized to V_0 but it would clarify the measured quantities for the readers. Furthermore, the "Methods" section should be widely improved and include at least the definitions of Vis , V_0 and TMZ .

The reviewers are right to mention that this definition is not given. We will add it in the main text, detailing that the normalization with $I_0/2$ is due to the fact that in the $\nu_L = -1 / \nu_R = +2$ configuration, there is a spectator edge state that does not participate to the interference. Indeed the interfering edge states carry the same spin while the spectator edge state carries an opposite spin. The energy cost to flip the spin at 9T is given by the Zeeman energy: $g\mu_B = 1\text{mV}$ that is much larger than the electronic temperature ($T=14\text{mK}$ what corresponds to $V=k_B T/e=1.2\mu\text{V}$). Therefore, the spectator edge channel is fully reflected meaning that half of the current cannot be transmitted. We will detail this effect in the main text.

Since the visibility is given by $Vis = (T_{MZ,max} - T_{MZ,min}) / (T_{MZ,max} + T_{MZ,min})$, the factor of renormalization $(I_0/2)$ or $(I_0/4)$ should not change its value. But it is true that the definition of the MZI transmission will change for the $\nu_L = -1 / \nu_R = +4$ configuration. There we assume that the two additional edge states are all reflected and $T_{MZ} = I_T / (I_0/4)$. We will detail this difference in the main text and define properly the definitions of Vis , V_0 and TMZ in the Methods.

3) Two middle-size interferometers are possible given the sample geometry (with a beam splitter at the first or the second side gate), why did the authors chose to discuss only one? Valley scatterers must exist in the bulk on both sides since the authors can form the small interferometer. The "universality" of the scaling shown in Fig.2b would benefit from a fourth example with same length as the presented intermediate.

The Reviewers are perfectly correct. We add these measurements in the main text showing that the intermediate interferometers with a beam splitter at the first or the second side gate give the same dependence of the visibility as a function of the temperature (see in Figure 2).

4) Did the authors also check the opposite polarity (i.e. holes in the left part and electrons on the right, like (2,-1) configuration?). And same question for the (-2,1) configuration? If not could the authors just comment on what prevented these explorations in the experiment? Given the different distances from the gates the claims may benefit from these additional data if they support the authors interpretation.

These additional configurations would be indeed very interesting. Unfortunately for the (2,-1), it is necessary to change the whole set up: the injection would be at the lower left ohmic contact and the amplifier at the top right ohmic contact. We have studied the MZI visibility for (2,-1) and (2,-2)

without doing a detailed study of its temperature dependence. For us, it was really important to measure the visibility dependence with the temperature in the same experimental configuration. We add these data in the supplementary. We have not investigated much the (-2,1) configuration.

5) Line 106-107, it is argued that the 3 interferometers “share the same PN interface, hence, the same properties of electron velocity and interaction strengths.” But the additional length of the intermediate and large interferometers have different gate configurations due to the presence of the side gates, closer to the edge channels than the top gate. According to the authors themselves, the distance of the metallic gate to the channels is of crucial importance in the mitigation of interactions through screening. We then respectfully disagree with this sentence, and believe that the “length” of the different interferometers shall not be the only thing to be taken into account.

In a recent theoretical paper [arXiv:2201.12025], we show that the distance between the interfering edge states is given by the electrostatics and the geometry. We demonstrate that the Landauer Buttiker picture cannot properly explain the physics of the PN junction. Instead we extend the Chklovski, Shklovskii and Glazman (CSG) model to graphene and given the distance between the graphene layer and the bottom/top gate, we extract quantitatively the distance between interfering edge channels without adjustable parameters. We consider two cases that we can compare with our experiments: In the configuration without side gates, the numerical simulations give a distance of 90 nm in the good agreement with the experimental value 83 nm (estimated from the Aharnov-Bohm period). In the configuration with the side gates, the simulations give a distance of 102 nm similarly to the experimental value 110 nm. This implies that the presence of the side gates modifies the distance between the interfering edge states by 10-20%. Hence it is reasonable to compare the interferometers based on their relative arm length difference as the first approximation, ignoring the possible modification by the side gates.

We agree with the Reviewer that this approximation should be explicitly mentioned in the main text. We have rephrased this sentence and have detailed the role of the side gates.

In addition, we would like to point out that the 10-20% modification cannot explain our experimental data. We revise Supplementary Material to include Fig. S14, in which we computed the thermal decay of the visibility for the $(\nu_n, \nu_p) = (2, -1)$ configuration with varying the strengths of the inter-channel interaction between the additional channel (having the opposite spin to the interfering electrons) and interferometer arms. We find that no decoherence happens when the additional channel interacts with the two arms equally. Instead, to fit the thermal visibility decay in Fig. 2 by the calculation based on the inter-channel interaction, it is required that one arm interacts with the additional channel about 10 times more strongly than the other arm. This asymmetry is unlikely in our setup, and cannot be obtained with the 10-20% modification of the interface edge channel separation by the side gates. This result also implies that the interchannel interaction is not the dominant decoherence source of our setup.

6) In the same vein, the comparison between the inter-channel distances and their distances to the gates is very confusing (lines 154-158 where it is not clear if it is the distance in the 2DEG plane or the vertical distance that is considered), so is the comparison with GaAs Mach-Zender interferometers. The reported distance variations are not very convincing (differing by roughly a factor 2-3), and may not be enough for a direct comparison: wouldn't the different dielectric constant of GaAs vs h-BN also play a noteworthy role in this discussion?

As shown in our recent theoretical paper[arXiv:2201.12025], the distance between the interfering edge states is essentially determined by the vertical distance to the gates with $W \propto d$ where W is the distance between the interfering edge state and d the distance to the gate. In GaAs MZI, the electron gas is always deeper than 100nm (to make the comparison clearer, we have added a detailed description of the sample with the different hBN thickness in the Supplementary material) As pointed out by the reviewers, we could also consider the dielectric constant of GaAs vs hBN. The dielectric constant of 10 layers of hBN is around 5 while it is 12 for GaAs. To observe a clear effect of the dielectric, we think that the dielectric should probably be much larger. In a recent experiment realized in the Sacepe's group [Helical quantum Hall phase in graphene on SrTiO₃, Science (2020)], a considerable dielectric constant (~ 10000) of the SrTiO₃ was necessary to screen Coulomb interaction. We add this discussion on the role of the dielectric in the supplementary.

7) In the main text, it is said that "Below 1 K, the visibility decays not exponentially, but algebraically, which means that thermal decoherence is suppressed". The link between the algebraic decay and the suppressed decoherence should be further justified. The article, in its present form, does not allow to establish this link properly (maybe this is just the word "suppressed" that should be replaced by "weak"). Furthermore, the algebraic and exponential decays are presented as two distinct regimes separated by a crossover temperature. However, these two behavior are captured by the same model detailed in the supplementary materials. Could the authors elaborate on that point, and pinpoint the physical meaning of the crossover temperature based on the model output?

We have plotted the visibility decay for the various interaction strengths in the Fig. S12. One can observe that the crossover occurs near the temperature of $T \sim \hbar v/k_B L$, and does not depend much on the interaction strength g . Physically, when an electron enters an interferometer arm, charge density fluctuations in the arm provide which-path information through the intrachannel capacitive interaction, reducing the interference. When the thermal width of the electron wavepacket is larger than the interferometer arm lengths, the fluctuation, and resulting dephasing is suppressed. We have added the discussion in the main text.

8) The model Hamiltonian sounds quite phenomenological, but no insight is given in the text concerning why this form is expected. We find it it rather crude for Nature communication. Could the authors give at least a hand-waving flavor for this Hamiltonian? And precise which gate does V_g refer to at the same time? Is it expected to be the same gate voltage for both sides or shall it be two different V_{gi} ? Besides, the model in the supplementary materials lacks references to be better justified.

The scaling behavior strongly restricts the possible dephasing mechanisms: The dephasing source should (1) be intrinsic to the interferometer (not by the extrinsic defects), and (2) have a length scale that is very large or very small compared to the interferometer sizes. Hence the scaling behavior implies that small charge puddles of the bulk are not the main decoherence source. The short-ranged inter-channel interactions satisfy the scaling behavior, but it is not a dominant decoherence source, as indicated by the other experimental results in Figs. 3 and 4 and the sample geometry. Among possible intra-channel interactions, it is well known in the Luttinger liquid theory that the short-ranged intra-channel Coulomb interaction merely renormalizes the propagation velocity of electrons.

Therefore, we consider a long-ranged intra-channel Coulomb interaction. Our model Hamiltonian in Eq. (1) is its simplest version, and it turned out that it is compatible to all of our findings and can cause the relevant dephasing. Basically, the Hamiltonian is parallel to the constant interaction model for Coulomb-blockade quantum dots. The reference electron number N_g is the charge capacitively determined by the gate voltages (by the top, bottom, and side gates with different capacitances) of the setup; in the previous version there was a typo --- N_g was misprinted by V_g . The Hamiltonian in Eq. (1) has been used in Ref. [5] and Refs. [S4, S5, S6] in Supplementary Material for analysis of the effect of the intra-channel interaction on the lobe pattern of GaAs Mach-Zehnder interferometers.

We have strengthened and clarified the discussion about the model in the main text and Methods, and Supplementary Materials. The scaling behavior of the model Hamiltonian in Eq. (1) have explicitly mentioned in the main text, and proved in Methods.

9) It is hard to understand, besides the geometrical argument of the gate distances, what is the specificity of graphene in this peculiar thermal behavior? We could draw no clear explanation from the text, which would give a feeling of the reason for this intra-channel dominating. In particular, we don't really understand why the authors compare their results only with GaAs-Mach-Zehnder interferometers (including a lot of self-citations), while they don't discuss recent results in graphene Fabry-Pérot interferometers (e.g. Déprez Nat. Nano. 2021, Ronen Nat. Nano. 2021). It could strengthen the paper if differences were discussed and if conclusions can be drawn on why the interferometers cited above do not reach this intra-channel dominated decoherence. Is it only a matter of length? If so are such interferometers also expected to reach this regime if designed smaller?

In two recent papers, graphene Fabry-Pérot interferometers are studied (e.g. Déprez Nat. Nano. 2021, Ronen Nat. Nano. 2021). In the Déprez Nat. Nano. 2021, the temperature dependence of the visibility is shown. It does not saturate at low temperature and an exponential dependence is found on the whole temperature range. For the large interferometer, oscillations are suppressed below 100mK. The presence of etch defined edges in the latter interferometer could explain this behavior as discussed in Ronen Nat. Nano. 2021 (see in Fig. 3b). In a gate defined interferometer, Ronen et al. extract a coherence length of 8.1 μ m at 32mK. In the present graphene MZI a coherence length of 1.24 μ m at 1K is found that is 4.78 times larger than in the Fabry Pérot quantum Hall interferometer. Ronen et al. do not observe any saturation of the visibility at low temperature (see in Fig. 1e). A larger interferometer together with a smaller coherence length may explain the absence of visibility saturation at low temperature in a graphene Fabry-Pérot. We add this discussion in the main text of the paper.

10) Other possible mechanisms for decoherence are not ruled out. In particular, we naively think of additional scattering events between the two arms of the interferometer itself. If intervalley scatterers do exist in the bulk (a pre-requisite need for observing interferences in the intermediate and small interferometers), could they also couple the co-propagating channels across the pn interface? What would then be the effect on the coherence?

Our experimental data do not indicate bulk scattering events (such as intervalley mixers) between the two arms. Such events make the interferometer size effectively be smaller, or result in beating in the interference pattern that shows multiple Aharonov-Bohm periods. We did not observe a clear signature of the beating. There may be such intervalley mixers in the bulk, but it may be sufficiently local so that they cannot mix the interferometer arms spatially separated by ~ 100 nm (estimated from the Aharonov-Bohm period).

11) Could the author discuss at some point in the text what limits V_0 to a non-unitary value? Is it only the coupling coefficients at the ends of the interferometers as in standard MZ interferometry or could the effect pointed in question 10 play a role in this finite visibility? Once again from Ref.2 insights can be obtained on the role of T_1 in the visibility but nothing is recalled in the present manuscript, and unless the paper is addressed only to MZ specialists it could be worth recalling it in one or two simple sentences.

In the ideal case, a visibility of 100% should be obtained when both beam splitters are tuned at $T=0.5$. In reference 2. , a precise study of the transmission dependence of the MZ visibility is done that clearly demonstrates the $\sqrt{T(1-T)}$ dependence of the visibility. Nevertheless, it is also shown that the maximum visibility is 60%. It remains unclear why we could not reach 100%. We have added a note on this point in the Methods.

12) A naïve question related to question 11: could temperature have a role on the intervalley scattering rate? If not could the authors unambiguously rule this out such that no doubt can remain on this effect contribution to visibility?

A temperature dependence of the transmission will affect the average MZI transmission. We do not observe such an effect and this hypothesis can be unambiguously ruled out. The temperature of our interest may be smaller than the energy scale for intervalley scattering; this is natural, since the intervalley scattering requires potential variation within a very short length scale (e.g. atomic distance) which means a very large energy scale. We add a note with the corresponding data in the main text (see in Figure 1).

13) Concerning the two models of intra and inter-channels interaction, we find the discussion very imbalanced in the favor of the scenario the authors want to demonstrate (namely intra-channel interaction dominating decoherence). For intra-channel model they apply an phenomenological decoherence with source-drain by multiplying the visibility by $\exp(-V^2/V_0^2)$, which makes Fig. S8B looking very much like the experimental data. On the other hand, they apply another formalism to capture the decoherence (ref. 7 in the supplementary) for the second model (inter-channel), which makes its results plotted in Fig. S9B looking very different from the experimental data (if we are not mistaking). Could the authors apply the same formalism to both models before comparing the results?

The reviewer suggests the very valid point for fair comparison between the models. Following the suggestion, we have added a new figure in Supplementary Material (Fig. S15) that compares the two models (Figs. S13 and S15), applying the same formalism of multiplying the visibility by $\exp(-V^2/V_0^2)$. We find that the inter-channel interaction model (combined with the exponential factor $\exp(-V^2/V_0^2)$) still cannot explain the features of the observed lobe pattern.

14) In the supplementary materials, we believe the scales are wrong in Figs. S3C, S4B, S4C, S5B and S5C, and that the authors meant V instead of mV .

We have corrected the scales of all the figures. We thank the Reviewers for their careful reading of the figures.

15) The conclusion of the paper is very elusive and projects the reader to very speculative applications regarding the paper's core (except maybe valleytronics), while not at all summing up and clarifying the results. A real conclusion on the findings of the paper would be appreciable.

We feel sorry for this. We revise the main text to include a proper conclusion on the findings of the paper.

REVIEWERS' COMMENTS

Reviewer #1 (Remarks to the Author):

The authors have addressed my comments in a satisfactory way. I am happy to support publication of the new version of the paper.

Reviewer #2 (Remarks to the Author):

We are pleased to see that the authors took the time required to adequately address the different concerns we raised, together with the ones of referee #1 which we all found relevant.

Overall, the manuscript has improved, and we feel it is now more suited for the broad audience of Nature Communication. We now recommend this manuscript for publication.